# Paraoxonase 1 Phenotype and Protein N-Homocysteinylation in Patients with Rheumatoid Arthritis: Implications for Cardiovascular Disease

**DOI:** 10.3390/antiox9090899

**Published:** 2020-09-21

**Authors:** Jolanta Parada-Turska, Grażyna Wójcicka, Jerzy Beltowski

**Affiliations:** 1Department of Rheumatology and Connective Tissue Diseases, Medical University, 20-090 Lublin, Poland; jolanta.parada@gmail.com; 2Department of Pathophysiology, Medical University, 20-090 Lublin, Poland; grazyna.wojcicka@umlub.pl

**Keywords:** paraoxonase 1, homocysteine, homocysteine thiolactone, protein N-homocysteinylation, rheumatoid arthritis, myeloperoxidase, atherosclerosis

## Abstract

Paraoxonase 1 (PON1) is the high density lipoprotein-associated esterase which inhibits the development of atherosclerosis by metabolizing lipid peroxidation products as well as hydrolyzing proatherogenic metabolite of homocysteine (Hcy), Hcy thiolactone, which otherwise reacts with lysine groups of proteins, thus forming N-Hcy-protein in a process referred to as protein N-homocysteinylation. Rheumatoid arthritis (RA) is the chronic inflammatory autoimmune disease associated with increased risk of cardiovascular complications, but the underlying mechanisms are incompletely understood. We examined PON1 status and N-homocysteinylation of serum proteins in patients with RA. Blood was collected from 74 RA patients and 70 control subjects. PON1 activity was measured toward synthetic (paraoxon, phenyl acetate) and natural (Hcy thiolactone) substrates. PON1 protein concentration was measured by ELISA. Total Hcy as well as N-Hcy-protein were measured in serum as well. PON1 activity toward Hcy thiolactone was lower in RA patients than in control subjects which was accompanied by increased concentration of N-Hcy-protein despite normal total Hcy concentration. PON1 protein concentration was unchanged in the RA group, but the specific enzyme activity was reduced. When RA patients were categorized according to the DAS28-ESR score, PON1 concentration and enzymatic activity were lower whereas N-Hcy-protein was higher in those with high disease activity. PON1 activity and Hcy thiolactone were correlated with DAS28-ESR score and myeloperoxidase concentration. In conclusion, RA is associated with deficiency of PON1 activity and increased protein N-homocyseinylation which may contribute to accelerated development of cardiovascular diseases.

## 1. Introduction

Rheumatoid arthritis (RA) is a chronic, systemic autoimmune disease characterized by symmetrical inflammatory polyarthritis of peripheral joints, leading to progressive irreversible erosive joint damage. It results in joint deformity, progressive physical disability, and impaired quality of life. RA affects approximately 1% of the adult population, most commonly people aged 40–70 years, with females being more likely to be affected than males [1].

The mortality in RA patients is around 1.5 times higher than in general population [2]. Accumulating evidence indicates that patients with RA have an increased risk of premature atherosclerosis and cardiovascular diseases (CVD) [3]. The main causes of increased cardiovascular risk in RA patients are inflammatory heart and vascular lesions accounted for by cytokines such as TNF and IL-6, endothelial dysfunction, oxidative stress, accelerated progression of atherosclerosis, and side effects of drug therapy, e.g., non-steroidal anti-inflammatory drugs and glucocorticoids as well as reduced physical activity due to joint lesions [4]. Presence of the traditional cardiovascular (CV) risk factors does not explain the higher rate of CV events in this group of patients, and currently RA is regarded as an independent risk factor for CVD [3,5].

Paraoxonase 1 (PON1) is a calcium-dependent esterase synthesized in the liver and circulating in the blood attached to high-density lipoproteins (HDL). PON1 protects plasma lipoproteins including low-density lipoproteins (LDL) and HDL themselves from oxidative modification by decomposing lipid peroxidation products such as fatty acid lactones [6]. In addition, PON1 hydrolyzes homocysteine (Hcy) thiolactone both in vitro [7] and in vivo [8,9]. Hcy thiolactone is the cyclic thioester of homocysteine which is, to a large extent, responsible for proatherogenic effect of this aminoacid. Hcy thiolactone reacts with ε-NH_2_ groups of protein lysine residues forming N-Hcy-protein in a process referred to as protein N-homocysteinylation [10]. PON1 knockout mice are more sensitive to atherosclerosis [11] and Hcy thiolactone-induced neurotoxicity [9]. In humans, low PON1 activity is the independent risk factor of acute CV events and is associated with most traditional risk factors of atherosclerosis such as hypercholesterolemia, diabetes, and smoking [12].

Several studies have addressed the relationship between PON1 and RA [13,14,15,16]. In some of them, only genetic polymorphism of PON1 (192RQ) has been examined [13]. If PON1 activity was measured, only synthetic substrates, usually paraoxon, were used. PON1 activity toward paraoxon is strongly affected by 192RQ polymorphism with QQ, QR, and RR genotypes being associated with low, intermediate, and high enzyme activity, respectively [12]. Consequently, PON1 activity toward paraoxon exhibits trimodal distribution and statistical significance of the differences is difficult to be demonstrated until large groups of patients are examined. In neither of previous studies PON1 activity toward one of its natural substrate, Hcy thiolactone, was measured.

In the present study, we aimed to address PON1 phenotype in RA patients more comprehensively. We measured PON1 polymorphism, enzyme activity toward synthetic substrates and Hcy thiolactone as well as enzyme concentration. In addition, the level of serum protein N-homocysteinylation and the Hcy thiolactone-dependent posttranslational modification were measured.

## 2. Materials and Methods

### 2.1. Patients and Control Subjects

The study was performed in 74 patients with clinical manifestation of RA (60 females, 14 males) treated in the Department of Rheumatology and Connective Tissue Diseases, Medical University, Lublin, Poland between January 2015 and December 2016. All patients fulfilled the American College of Rheumatology (ACR) 1987 revised criteria for RA diagnosis [17]. RA patients were classified according to disease activity score (DAS) using 28 joint count and the erythrocyte sedimentation rate (ESR) (DAS28-ESR) [18]. Remission, low activity, moderate activity, and high activity of RA were defined as DAS28-ESR < 2.6 (*N* = 7), 2.6–3.2 (*N* = 8), 3.2–5.1 (*N* = 17), and >5.1 (*N* = 42), respectively. Considering non-biological conventional synthetic disease-modifying antirheumatic drugs (DMARDs), 60 patients were treated with methotrexate (only methotrexate *n* = 47, methotrexate with chloroquine n = 10, methotrexate with other drugs *n* = 3). Regarding immunosuppressive drugs, two patients received azathioprine and an additional two received azathioprine and cyclosporine. Forty-four patients received biological DMARDs such as adalimumab, certolizumab, etanercept, golimumab, infliximab, tocilizumab, and rituximab. Fifty-four patients with RA were treated with low-dose of oral glucocorticosteroids, mostly methylprednisolone (*n* = 41) or prednisone (*n* = 13). In addition, 70 sex- and age-matched healthy control subjects (58 females and 12 males) were included as the control group. Written informed consent was obtained from all participants and the study was approved by the Ethics Committee of the Lublin Medical University (the approval number: KE-0254/163/2011) and have therefore been performed in accordance with the ethical standards laid down in the 1975 Declaration of Helsinki.

Blood samples were withdrawn from the antecubital vein after the overnight fast. Serum and EDTA plasma were prepared from venous blood, aliquoted in 1.5 mL Eppendorf tubes and stored at −80 °C. Each sample was thawed only once before the analysis.

### 2.2. Anthropometric, Clinical, and Laboratory Data

Hemoglobin concentration, complete blood count, plasma protein concentration, alanine (ALT) and aspartate (AST) aminotransferases, erythrocyte sedimentation rate (ESR), total cholesterol, HDL-cholesterol, triglycerides, and C-reactive protein (CRP) were measured by routine clinical laboratory methods. Non-HDL cholesterol was calculated as the difference between total and HDL-cholesterol and LDL-cholesterol was calculated by the Friedewald formula.

Apolipoprotein A-I was measured using ELISA kit (Abcam, cat. #ab108804). The assay range was 0.625–20 μg/mL, the intra- and interassay coefficients of variation (CVs) were 4.9% and 7.2%, respectively. Apolipoprotein A-II was assayed by Abcam fluorescence-based ELISA kit (cat.no. ab229423). The sensitivity of the assay, the intra-assay CV and the inter-assay CV were 7.2 pg/mL, 2.2% and 8.1%, respectively. Myeloperoxidase (MPO) concentration was measured by Human Myeloperoxidase ELISA kit (cat.no. Ab119605, Abcam, Cambridge, MA, USA).

### 2.3. PON1 Activity Toward Synthetic Substrates

PON1 activity toward paraoxon was determined by measuring the initial rate of p-nitrophenol formation in the assay mixture containing 2.0 mM paraoxon and 2.0 nM CaCl_2_ in 800  μL of 100  mM Tris–HCl buffer (pH 8.0) at a temperature of 37 °C [19]. Paraoxon stock solution was prepared in the air extraction fume hood by the operator wearing face mask and nitrile gloves to prevent from accidental contact or inhalation. After the assay, waste glass, tubes, and pipettes were treated with 1 M NaOH to accelerate paraoxon hydrolysis. For the assay, 20 µL of serum was added and absorbance was read at 412 nm for 2  min. The blank sample containing assay mixture without serum was assayed simultaneously to correct for spontaneous substrate hydrolysis. Increase in absorbance of the blank sample was subtracted from the increase in absorbance of the test sample and enzyme activity was calculated from E_412_ of p-nitrophenol (18 290  M^−1^ cm^−1^) and was expressed in U/mL; one unit of PON1 hydrolyzes 1 nmol of paraoxon/min.

PON1 activity toward phenyl acetate (arylesterase activity) was determined by measuring the initial rate of phenol formation, monitored at 270  nm, in the assay mixture (3  mL) containing 2  mM phenyl acetate, 2  mM CaCl_2_, and 10  μL serum in 100  mM Tris–HCl (pH 8.0). Increase in absorbance at 270  nm was monitored for 3  min and the activity was calculated from *E*_270_ = 1310  M^−1^ cm^−1^). The results are expressed in U/mL, 1 U arylesterase hydrolyses, 1  μmol phenyl acetate per min [20].

### 2.4. PON1 Polymorphism

PON1 exhibits Q192R polymorphism (glutamine to arginine substitution) which affects enzyme activity toward some organophosphate compounds; QQ, QR, and RR genotypes. PON1 phenotyping was performed by the dual substrate methods of Eckerson and collegues [21]. PON1 activity toward paraoxon was additionally measured in a 50 mM glycine buffer (pH 10.5) in the presence of 1 M NaCl. The rate of salt-stimulated PON1 activity toward paraoxon to arylesterase activity was calculated for each sample. Patients with the ratio <2.5, 2.6–7.5, and >7.5 were classified as QQ, QR, and RR.

### 2.5. PON1 Activity Toward Homocysteine Thiolactone

PON1 activity toward Hcy-thiolactone was determined by measuring the amount of hydrogen ions generated during thiolactone hydrolysis to Hcy in the assay mixture (800  μL) containing 5  mM substrate, 1  mM CaCl_2_, 0.0005% bovine serum albumin, 0.004% phenyl red, and 20 μL of serum in 5  mM HEPES buffer. Although neutral pH is not optimal for PON1, Hcy thiolactone is unstable in alkaline conditions and therefore enzyme activity toward Hcy thiolactone was measured at pH 7.0 [19]. The absorbance was monitored for 4 min at 412 nm, and the results were expressed in nmol min^−1^ mL^−1^. The blank sample containing incubation mixture without serum was simultaneously assayed to correct for spontaneous substrate breakdown. PON1 activity toward Hcy thiolactone was calculated from the standard curve prepared by titrating the assay medium with the different HCl concentrations.

### 2.6. PON1 Protein Concentration

The concentration of PON1 was measured in serum using Paraoxonase 1 Human ELISA kit (Cat. No.: RD191279200R, BioVendor, Brno, Czech Republic).

### 2.7. Serum Homocysteine Concentration and Protein N-Homocysteinylation

Serum total Hcy was assayed by enzyme immunoassay using commercially available kit based on the methods proposed by the manufacturer (Axis Shield Diagnostics Ltd., Dundee, UK). In this method, Hcy bound to serum proteins by disulfide bonds is first released by adding dithiotreitol (DTT) and then converted enzymatically to *S*-adenosylhomocysteine (SAH) which is detected by specific antibodies. The detection limit of the assay is 0.5  μM, whereas the intra- and inter-assay CV values were 6% and 9%, respectively.

To measure the amount of N-Hcy-protein, after DTT-induced liberation of disulfide-bound Hcy serum proteins were precipitated with ethanol and hydrolyzed at 110 °C in the presence of 6 M HCl. Hydrolyzate was evaporated under N_2_ and Hcy thiolactone was converted to Hcy by adding NaOH. Hcy was then assayed by the method described above [19].

### 2.8. Reagents

Unless otherwise stated, all the reagents were obtained from Sigma-Aldrich (Steinheim, Germany).

### 2.9. Statistical Analysis

The results are expressed as mean ± SD. The results obtained in control and RA groups as well as in subgroups of RA patients categorized according to the disease activity were compared by two-tailed Student *t*-test. One-way analysis of variance with Tukey post-hoc test was used to analyze data in three different groups (control subjects, RA patients with remission-to-moderate activity, and RA patients with high disease activity). The distribution of PON1 genotypes was analyzed by chi-square test. Correlations between variables was analyzed by calculating Person’s “r” coefficient. To identify factors independently associated with PON1 activity toward Hcy thiolactone and N-Hcy-protein, we performed multiple regression analysis with total Hcy, HDL-cholesterol, apolipoprotein A-I, apolipoprotein A-II, DAS28-ESR, CRP, MPO, treatment with methotrexate and treatment with biological DMARDs as the putative independent variables. Binary variables were used for methotrexate and biological DMARDs (0—no, 1—yes). *p* < 0.05 was considered significant in all analyses.

## 3. Results

### 3.1. Characteristics of RA Patients and Control Subjects

There was no difference in age, body weight, body mass index (BMI), waist circumference, hip circumference, and waist-to-hip ratio between RA patients and control subjects (Table 1). Disease duration in RA patients was 3–420 months (median/interquartile range: 134/72–219 months) (Table 1). Hemoglobin concentration, red blood cell count, mean red blood cell volume, platelet count, white blood cell count, and total plasma protein concentration were similar in both groups as well (Table 2). Estimated glomerular filtration rate (eGFR) was normal (>90 mL/min) in 50 (67.6%) and 54 (77.1%) of RA patients and control subjects, respectively. Stage 2 (eGFR = 60–89 mL/min) and stage 3a (eGFR = 45–59 mL/min) of chronic kidney disease (CKD) were recognized in 21 (28.4%) and 3 (4.1%) patients with RA. Among control subjects, stage 2 CKD was recognized in 16 subjects and no subjects suffered from CKD stage 3a or higher. Type 2 diabetes and corticosteroid-induced diabetes were recognized in three and two RA patients, respectively. Thirty-nine RA patients had arterial hypertension and two were after myocardial infarction. No cases of arterial hypertension or ischemic heart disease were included in the control group. Aspartate and alanine aminotransferase activities did not differ between RA patients and healthy controls.

Total cholesterol, LDL-cholesterol as well as triglycerides tended to be lower in RA patients than in control group but the difference was not significant. HDL-cholesterol was not different between groups. Similarly, apolipoprotein A-I and A-II concentrations did not differ between groups (Table 2). As expected, ESR, CRP, and MPO concentrations were much higher in RA patients than in control subjects.

### 3.2. PON1 Activities, Protein Concentrations, and Phenotype

QQ was the most frequent phenotype in both control and RA groups followed by QR and the least frequent RR. There was no difference in PON1 phenotype distribution between both groups (Appendix A).

PON1 activities toward phenyl acetate and Hcy thiolactone were lower in RA patients than in control subjects by 39.9% and 35.9%, respectively (Figure 1a). PON1 activity toward paraoxon was also lower in subgroups of RA patients with each of three phenotypes (Figure 1b). However, PON1 concentration was similar in RA and control groups (22.2 ± 3.2 µg/mL and 24.8 ± 3.5 µg/mL, respectively). Specific PON1 activity, that is the ratio between enzyme activity and concentration, was reduced in RA patients when either phenyl acetate or Hcy thiolactone were used as the substrates. Interestingly, specific PON1 activity toward paraoxon was reduced in RA patients with QQ and QR phenotypes, but in those with RR phenotype (Appendix A)

### 3.3. Plasma Homocysteine and N-Hcy-Protein

Total plasma Hcy was similar in both groups. In contrast, N-Hcy-protein was 18.8% higher in RA patients than in control subjects (Figure 2).

### 3.4. PON1 Status, Homocysteine, and N-Hcy Protein in RA Patients Stratified According to Disease Activity

Next, RA patients were divided into two subgroups with remission-to-moderate disease activity (DAS28-ESR ≤ 5.1, *n* = 32) and high disease activity (DAS28-ESR > 5.1, *n* = 42). PON1 phenotype distribution did not differ between groups. PON1 protein concentration was 32.1% lower in the subgroup with high disease activity. When both patients’ subgroups were compared with control group by ANOVA, PON1 concentration was significantly reduced only in patients with high disease activity (−25.0%, *p* < 0.01). PON1 activity toward Hcy thiolactone was lower in patients with high than in those with remission-to-moderate activity (Table 3). Specific enzyme activity toward Hcy thiolactone was similar in both RA subgroups but in each of them was lower than in control group when compared by ANOVA (remission-to-moderate activity: −25.6%, *p* < 0.01; high activity: −35.7%, *p* < 0.001). Total plasma Hcy was similar irrespectively of the disease activity; however, N-Hcy-protein was by 20.2% higher in patients with high than in those with remission-to-moderate disease activity (Table 3). N-Hcy-protein was higher in patients with high disease activity in comparison to control subjects by 30.5% (*p* < 0.01). However, there was no difference in N-Hcy-protein between control subjects and patients with remission-to-moderate disease activity.

### 3.5. PON1 Status, Homocysteine, and N-Hcy Protein in RA Patients and Control Subjects with Normal Renal Function

Chronic kidney disease is associated with PON1 deficiency and hyperhomocysteinemia [22]. To take into consideration the possible contribution of renal impairment to low PON1 status in RA patients, we performed the separate analysis in subgroups of control subjects and RA patients with normal renal function (eGFR > 90 mL/min). As shown in Appendix A, the distribution of PON1 phenotypes as well as enzyme concentration did not differ between groups. PON1 activities toward phenyl acetate and Hcy thiolactone were lower whereas N-Hcy-protein was higher in RA patients than in healthy subjects.

### 3.6. Lipid Profile, PON1 Status, Homocysteine, and N-Hcy Protein in Healthy and RA Females Startified According to Menopausal Status

Estrogens have beneficial effect on plasma lipids, in particular increase HDL, and also have antioxidant activity. Therefore, we examined if menopause has any effects on blood lipids, PON1 and homocysteine. Most females in the control and RA groups were after the menopause (82.8% and 65.0%, respectively). Both control and RA females after the menopause were significantly older than premenopausal ones. In the control group, mean total cholesterol, LDL-cholesterol and non-HDL cholesterol as well as triglycerides were higher in the postmenopausal subgroup than in premenopausal subgroup, but the difference was statistically significant only for total cholesterol. PON1 concentration and activity as well as total Hcy and N-Hcy protein did not differ between healthy pre- and postmenopausal females (Appendix A).

In females with rheumatoid arthritis, there was no difference in lipid levels between pre- and postmenopausal subgroups (Appendix A). Among disease activity markers, CRP level was significantly higher in females after the menopause whereas ESR and DAS28-ESR did not differ between both subgroups. There was no difference in PON1 concentration, PON1 activity, and N-Hcy protein between pre- and postmenopausal females; however, total Hcy was slightly but significantly higher in the postmenopausal subgroup.

### 3.7. Correlations

In the control group, PON1 activity toward Hcy thiolactone was positively correlated with enzyme’s activity toward phenyl acetate (*r* = 0.53, *p* < 0.5). No correlation was observed between either of these activities and N-Hcy-protein, HDL-cholesterol, apolipoprotein A-I, or apolipoprotein A-II concentrations. In addition, N-Hcy-protein was not correlated with total Hcy concentration (not shown).

In the RA patients, PON1 activity toward Hcy thiolactone was negatively correlated with DAS28-ESR (*r* = −0.78, *p* < 0.01), CRP (*r* = −0.52, *p* < 0.05), and MPO (*r* = −0.69, *p* < 0.01) but not with total Hcy, HDL-cholesterol, apolipoprotein A-I, apolipoprotein A-II, or eGFR. The level of N-Hcy-protein was positively correlated with total Hcy (*r* = 0.38, *p* < 0.05), DAS28-ESR (*r* = 0.48, *p* < 0.05), and MPO (*r* = 0.54, *p* < 0.05) and negatively with PON1 activity toward Hcy thiolactone (*r* = −0.52, *p* < 0.05).

Finally, multiple regression analysis was performed to identify factors independently associated with PON1 activity toward Hcy thiolactone and N-Hcy-protein in patients with RA. DAS28-ESR (β = -0.33, *p* < 0.05) and MPO level (β = −0.30, *p* < 0.05) but not total Hcy, CRP, HDL-C, eGFR, current methotrexate use, or current use of biological DMARDs were negatively associated with PON1 activity and positively with N-Hcy-protein (DAS28-ESR: β = 0.43, *p* < 0.05; MPO level: β = 0.36, *p* < 0.05).

## 4. Discussion

The main findings of this study are that: (1) plasma lipid profile did not differ significantly between RA patients and control subjects, (2) distribution of PON1 phenotypes was similar in RA and control groups, (3) PON1 activity is lower in RA patients, (4) total Hcy level is similar in RA patients and control subjects, (5) N-Hcy-protein is higher in RA, (6) both PON1 activity and Hcy thiolactone are associated with disease activity and MPO concentration.

The assessment of the effect exerted by RA on plasma lipoprotein metabolism is difficult. Some studies have demonstrated that total cholesterol, LDL-cholesterol, HDL-cholesterol, and/or triglycerides are lower in RA patients than in control subjects [23]. It is postulated that RA-associated hypolipidemia may result from chronic inflammation-induced hypercatabolism [24]. On the other hand, other studies showed no difference in plasma lipoprotein parameters between RA patients and control subjects [25]. It has been demonstrated that DMARDs increase plasma lipid levels [24]. In the present study, most patients were treated with non-biological and/or biological DMARDs which could increase lipid levels thus explaining no significant difference in parameters reflecting lipoprotein metabolism in RA in comparison to control group.

Few studies have demonstrated the significant relationship between Q192R polymorphism of PON1 and the prevalence of RA [26]; however, in other studies no such relationship was observed [13,27]. In addition, according to the recent meta-analysis there is no correlation between this polymorphism and RA susceptibility [28]. Many authors focused on the role of this polymorphism in the susceptibility to CV and other diseases [29,30]. However, according to other studies, PON1 activity and concentration are more important than the genotype [31]. Although Q192R polymorphism affects enzyme’s activity toward some synthetic organophosphate compounds such as paraoxon or diazoxon, it has no effect on other activities including lipolactonase activity which is crucial for protection against oxidative stress [32]. Anyway, the present study supports previous results indicating that Q192R polymorphism is not associated with RA susceptibility.

Several studies have demonstrated that PON1 activity is reduced in patients with RA in comparison to healthy subjects which is accompanied by the increase in lipid peroxidation products and/or other markers of oxidative stress [13,14,15,27]. Our results fully support these data. Moreover, we demonstrated for the first time that serum PON1 activity toward its natural substrate, Hcy thiolactone, is also reduced in RA. The mechanism of PON1 deficiency in RA patients is not clear at present. According to many studies, PON1 level is positively correlated with HDL and/or apolipoprotein A-I for review, see [12,32]. PON1 protein is taken up from the liver by HDL and is transported attached to these lipoproteins. In addition, apolipoprotein A-I is crucial for PON1 binding and stability [33]. However, we did not find any differences in either HDL-cholesterol or apolipoprotein A-I between RA and control groups. Nevertheless, PON1 activity is affected also by lipid components of HDL [34]. Although HDL-cholesterol is inversely associated with CV diseases, medications which increase HDL failed to reduce mortality in clinical trials [35]. HDL-cholesterol concentration is only one among many factors which affect HDL functions such as reverse cholesterol transport or antioxidant properties. Indeed, in many pathological conditions HDL functionality may be impaired despite normal concentration. Chronic inflammatory diseases including RA are well-known to affect structure of these lipoproteins and to impair their functionality [36]. Thus, we cannot exclude the possibility that changes in lipid constituents of HDL could affect PON1 activity. Indeed, RA is associated with lower cholesterol ester and higher free cholesterol and triglyceride content in HDL [37]. Whether these or other lipid alterations affect PON1 activity in RA remains to be established.

RA is associated with chronic activation of inflammatory cells and overproduction of proinflammatory cytokines such as IL-1β, TNF-α, IL-6, and others. It has been demonstrated that some of these cytokines such as IL-1β and TNF-α inhibit PON1 synthesis in cultured hepatocytes in vitro and in the liver in vivo [38]. In the present study, PON1 protein concentration was reduced only in patients with high RA activity suggesting that this mechanism could operate in this subgroup. However, in contrast to IL-1β and TNF-α, IL-6 has been demonstrated to stimulate PON1 expression [39]. Differential effects of various cytokines as well as treatment with anti-cytokine DMARDs could modify PON1 synthesis in RA thus contributing to different PON1 protein levels depending on the disease activity. Nevertheless, PON1 activity toward Hcy thiolactone correlated with DAS28-ESR but not with DMARDs use suggesting that disease activity rather than pharmacotherapy determines enzyme activity. It should be noted that, unlike protein concentration, specific PON1 activity was significantly reduced in subgroups with remission-to-moderate and high disease activities.

Decrease in PON1 concentration in high disease activity subgroup could be accounted for not only by its reduced synthesis in the liver but also by its displacement from HDL by serum amyloid A protein (SAA) [40] which is elevated in patients with RA [41]. PON1 is activated and stabilized by HDL-associated apolipoprotein A-I and lipid components whereas HDL-free enzyme is unstable and has low specific activity [33,34]. Thus, SAA-mediated displacement of PON1 from HDL may contribute to both reduced PON1 concentration and specific enzyme activity. PON1 is also inactivated by reactive oxygen species (ROS) and lipid peroxidation products [42]. Activated inflammatory cells in RA patients overproduce ROS which contribute to oxidative stress and tissue injury [43]. Negative correlation between PON1 activity and markers of oxidative stress such as lipid hydroperoxides or protein thiol groups observed in RA patients in some studies [14] supports this possibility.

Myeloperoxidase produced by neutrophils and, to a lesser extent, macrophages, plays an important role in host defense by producing highly bactericidal hypochlorous acid [44]. At least one study found increase in HDL-associated MPO in RA [45]. Remarkably, in the present study, MPO concentration was negatively correlated with PON1 activity in both uni- and multivariate analysis. Proinflammatory HDL isolated from patients with CVD contain more 3-chlorotyrosine and 3-nitrotyrosine; the specific products of MPO-dependent reactions [46]. In addition, although MPO substrate, H_2_O_2_, had no effect on apolipoprotein A-I function, chlorination of this apolipoprotein reduced the ability of HDL to take up cholesterol [47]. HDL-associated MPO was negatively correlated with PON1 activity in patients with type 2 diabetes and coexisting CVD, non-diabetic patients with chronic ischemic heart disease, and patients with acute coronary syndromes [48]; the results similar to that observed by us in RA patients. In addition, Deakin et al. have demonstrated that hypochlorite-induced oxidation of apolipoprotein A-I has no effect on PON1 uptake from hepatocytes but reduces enzyme activity in HDL more remarkably than oxidation induced by other ROS [49]. Together, these data raise the possibility that inhibition of PON1 may represent an additional mechanism by which MPO provokes HDL dysfunction in RA.

Hcy thiolactone is synthesized inside the cells following non-specific binding of Hcy by methionine tRNA synthetase. Instead of being incorporated into proteins, Hcy is then converted to Hcy thiolactone which specifically binds to ε-NH_2_ groups of protein lysine residues [50]. Remarkably, N-Hcy-protein is not included when ‘total’ Hcy is measured by most laboratory methods including that used by us because its measurement requires prior protein hydrolysis. Free Hcy thiolactone circulates at low nanomolar concentration; however, protein N-homocysteinylation is the irreversible posttranslational modification which accumulates over time resulting in micromolar concentrations of N-Hcy-protein. The level of free Hcy thiolactone as well as N-Hcy-protein is affected by factors such as Hcy concentration, activity of Hcy thiolactone degrading enzymes including PON1, and urinary excretion [50]. N-homocysteinylation was demonstrated as being important for several proteins in the pathogenesis of CVD such as angiotensin-converting enzyme, apolipoproteins, or fibrinogen [50]. For example, N-homocysteinylation of fibrinogen renders it more resistant to fibrinolysis which may contribute to pro-thrombotic tendency [51]. The extent of serum protein N-homocysteinylation as well as the level of free Hcy thiolactone in plasma and urine are increased in patients with ischemic heart disease or ischemic stroke [52,53]. In addition, N-homocysteinylated proteins are immunogenic and the relationship between CVD and the titer of specific antibodies was also reported [54]. Protein N-homocysteinylation results in the incorporation of the additional thiol group which may render target protein more susceptible to oxidation. Indeed, oxidative stress is involved in detrimental effects of Hcy thiolactone in different experimental systems [55,56,57,58]. Thus, decomposing Hcy thiolactone may contribute to antioxidant activity of PON1.

Herein, we demonstrate for the first time that protein N-homocysteinylation is enhanced in RA patients but only in those with high disease activity despite unchanged total Hcy. In addition, Hcy thiolactone was positively correlated with total Hcy and negatively with PON1 activity only in RA patients but not in healthy subjects. These data suggest that both Hcy level and PON1 activity become important determinants of protein N-homocysteinylation when Hcy thiolactone balance is stressed by factors such as PON1 deficiency.

## 5. Conclusions

RA is characterized by reduced PON1 activity toward its natural substrate, Hcy thiolactone, which results from both decrease in enzyme concentration and specific activity. PON1 deficiency does not result from the impairment of renal function or HDL/apolipoprotein A-I deficiency but is associated with high disease activity of RA and high MPO level. Protein N-homocysteinylation is increased in RA patients and may contribute to accelerated atherosclerosis and CV morbidity.

## Figures and Tables

**Figure 1 antioxidants-09-00899-f001:**
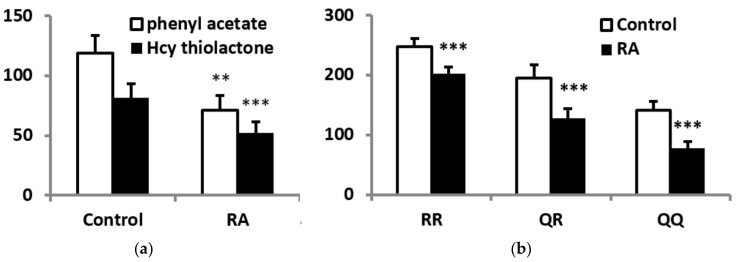
PON1 activity in RA patients and control subjects. (**a**) serum PON1 activity toward phenyl acetate (U/mL) and Hcy thiolactone (nmol × min^−1^ × mL). (**b**) PON1 activity toward paraoxon in RA patients and control subjects with various enzyme phenotypes. ** *p* < 0.01, *** *p* < 0.001 vs. control group.

**Figure 2 antioxidants-09-00899-f002:**
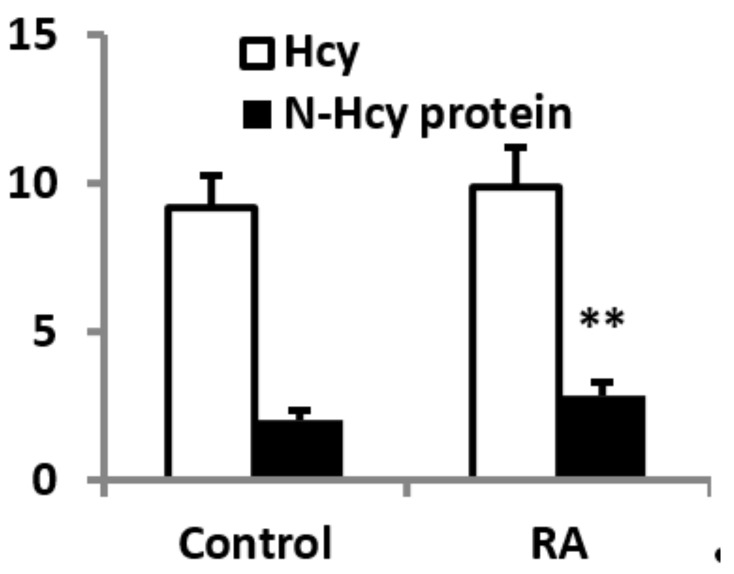
Total serum homocysteine (Hcy) and N-Hcy-protein (both in μM) in RA patients and control subjects. ** *p* < 0.01 vs. control group.

**Table 1 antioxidants-09-00899-t001:** Characteristics of RA patients and control subjects.

	Control (*n* = 70)	RA (*n* = 74)
Age (years)	52.3 ± 8.8	55.4 ± 12.2
Males/females	12/58	14/60
Body weight (kg)	62.1 ± 11.3	65.6 ± 14.4
Body mass index (kg/m2)	24.9 ± 12.0	26.4 ± 13.5
Waist circumference (cm)	82.3 ± 10.6	84.7 ± 13.8
Hip circumference (cm)	97.9 ± 8.2	99.6 ± 8.6
Waist-to-hip ratio	0.81 ± 0.10	0.84 ± 0.11
Disease duration (months)	-	151 ± 107

**Table 2 antioxidants-09-00899-t002:** Laboratory data of RA patients and control subjects.

	Control (*n* = 70)	RA (*n* = 74)
Hemoglobin (g/dL)	13.2 ± 1.8	12.3 ± 1.7
Red blood cell count (mln/μL)	4.51 ± 0.49	4.42 ± 0.53
MCV	88.7 ± 7.6	84.8 ± 8.3
Platelet count (10^3^/ μL)	336 ± 115	324 ± 120
White blood cells (10^3^/ μL)	7.23 ± 2.04	7.84 ± 2.47
Total plasma protein (g/dL)	7.23 ± 0.68	6.98 ± 0,59
AST (U/L)	22.1 ± 11.1	24.5 ± 13.3
ALT (U/L)	24.7 ± 11.3	26.1 ± 28.6
Total cholesterol (mg/dL)	211 ± 32	185 ± 46
LDL-cholesterol (mg/dL)	123 ± 21	106 ± 36
HDL-cholesterol (mg/dL)	54 ± 15	58 ± 18
Non-HDL cholesterol (mg/dL)	157 ± 28	128.3 ± 41.6
Triglycerides (mg/dL)	132 ± 25	108 ± 48
Apolipoprotein A-I (μg/mL)	1.12 ± 0.02	1.05 ± 0.03
Apolipoprotein A-II (μg/mL)	0.03 ± 0.01	0.04 ± 0.02
ESR (mm/1 h)	5.6 ± 3.8	35.4 ± 25.8 ***
CRP (mg/L)	3.34 ± 0.76	27.9 ± 31.9 ***
MPO (ng/mL)	1.5 ± 1.2	54.5 ± 13.2 ***

*** *p* < 0.001.

**Table 3 antioxidants-09-00899-t003:** PON1 status, Hcy, and N-Hcy protein in RA patients stratified according to the disease activity.

	Remission-to-Moderate Activity (*n* = 32)	High Activity (*n* = 42)
PON1 polymorphism		
QQ	17	22
QR	13	16
RR	2	4
PON1 concentration (μg/mL)	27.4 ± 3.2	18.6 ± 2.1 **
PON1 activity toward Hcy thiolactone (nmol × min^−1^ × mL^−1^)	57.2 ± 12.1	42.1 ± 10.2 **
Specific PON1 activity toward Hcy thiolactone (nmol × min^−1^ × μg^−1^)	2.11 ± 0.53	2.44 ± 0.61
Total Hcy (μM)	9.12 ± 0.97	10.26 ± 1.24
N-Hcy-protein (μM)	2.53 ± 1.04	3.04 ± 0.88 **

** *p* < 0.01 vs. remission-to-moderate activity subgroup.

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
