# Peer review of "Paraoxonase 1 Phenotype and Protein N-Homocysteinylation in Patients with Rheumatoid Arthritis: Implications for Cardiovascular Disease"

_antioxidants, 2020, doi:10.3390/antiox9090899_

Round 1
Reviewer 1 Report
Compared to current literature, this study provides additional insights into PON1 activity, beyond polymorphism, in patients with RA. I have only 2 comments:
- I am a little perplexed about the role of PON1. On one hand, I understand that it prevents oxidation of lipoproteins, which include LDL. On the other hand, as you have mentioned, it hydrolyses Hcy-thiolactone. How is its former role related to your current study and impact on CV risk in patients with RA?
- It seems unclear which patients was the regression analysis performed on? RA patients only or all patients?
Author Response
Comment 1:
I am a little perplexed about the role of PON1. On one hand, I understand that it prevents oxidation of lipoproteins, which include LDL. On the other hand, as you have mentioned, it hydrolyses Hcy-thiolactone. How is its former role related to your current study and impact on CV risk in patients with RA?
Response:
Thank you very much for raising this important issue. PON1 is the esterase active toward many natural and synthetic substrates. However, it is widely accepted now that PON1’s primary substrates are different natural and synthetic lactones. This hypothesis is supported by the fact that PON1 is structurally similar to many vertebrate, invertebrate and even bacterial lactonases. In addition, the related enzymes, PON2 and PON3, which most likely originated during the evolution by gene multiplication have also lactonase activities, although do not hydrolyze Hcy thiolactone. It is appreciated that antioxidant properties of PON1 are mainly related to hydrolysis of fatty acid lactones. During reactive oxygen species-induced lipid peroxidation, hydroperoxyl (-OOH) and hydroxyl (-OH) derivatives of fatty acids originate; the latter can form lactones by creating intramolecular ester bond between hydroxyl and carboxyl group thus generating internal 5- or 6-carbon ring. Thus, antioxidant (fatty acid lactonase) and Hcy thiolactonase activities of PON1 may be parallel but independent ones.
However, decomposing Hcy thiolactone may also contribute to antioxidant properties of PON1. Protein N-homocysteinylation results in the incorporation of addithional thiol (-SH) group into protein molecule. Thiol groups may enable intra- and intermolecular disulfide bond formation but also are susceptible to oxidation rendering proteins prone to reactive oxygen species. Several studies have demonstrated that Hcy thiolactone induces oxidative stress and antioxidants are protective against Hcy thiolactone-induced toxicity in various experimental system. This issue is briefly discussed in the revised manuscript (lines 388-392).
In previous studies in which PON1 genotype and/or activity were examined in patients with RA, the results were interpreted only in the context of its antioxidant properties, and negative correlation between PON1 and markers of oxidative stress was observed. Like in all inflammatory diseases, oxidative stress plays an important role in both joint injury in RA and complications affecting remote organs including cardiovascular system (lines 314-316 of the revised manuscript). However, in the present study we addressed for the first time the relationship between PON1, Hcy thiolactone and protein N-homocysteinylation in this disease.
Comment 2
It seems unclear which patients was the regression analysis performed on? RA patients only or all patients?
Response:
Univariate correlations were analyzed separately in the control and RA groups (lines 272-276 and 277-281, respectively). Multiple regression analysis was performed in RA patients to identify factors independently associated with PON1 activity and protein N-homocysteinylation. This information is included in the revised manuscript (line 283).
All changes made in the text are marked in red font in the revised manuscript. Additional references (7-9 and 55-58) are included in the reference list to support discussion suggested by the reviewers. The numbering of other references was updated accordingly in the text and the reference list. Finally, two additional supplementary tables (S4 and S5) were added to present results requested by one of the referees. In all responses we refer to line numbering in the revised manuscript.
Reviewer 2 Report
The article is certainly relevant and contains important and interesting results. However, the article needs some work:
My comments:
1. The authors do not indicate age limits for patients in the main and control groups. This is very important because some of the women in both groups were apparently at the pre-menopausal age, when they had sufficient levels of estrogen, which protect against oxidative processes.
2. It is important to present the obtained results additionally in the table in different subgroups (women before and after menopause). Perhaps, in this case, correlations and associations appear with some of the studied parameters, especially with HDL.
3. In the list of references, it is necessary to indicate DOI for many sources.
Author Response
Comment 1
The authors do not indicate age limits for patients in the main and control groups. This is very important because some of the women in both groups were apparently at the pre-menopausal age, when they had sufficient levels of estrogen, which protect against oxidative processes.
Comment 2
It is important to present the obtained results additionally in the table in different subgroups (women before and after menopause). Perhaps, in this case, correlations and associations appear with some of the studied parameters, especially with HDL.
Responses to comments #1 and #2
This is certainly a very important issue. Most females in both control and RA groups were after the menopause. We performed the additional analysis of the most important results according to menopausal status in control (Table S4) and RA (Table S5) females. The results of this analysis are described in the new section 3.6 in the revised version. No major effects of menopausal status on lipid profile and disease activity were observed. In addition, PON1 concentration and activity toward Hcy thiolactone as well as protein-bound Hcy thiolactone (now referred to as N Hcy protein) did not differ between pre- and postmenopausal subgroups. In the RA patients, but not in the control group, total Hcy was higher in postmenopausal females but this difference, although significant, was relatively small (below 20%).
The effect of menopause on blood lipids, PON1 and homocysteine was addressed in many studies and the results are very variable. Because this was not the primary aim of our study, we did not discuss the results presented in section 3.6 separately in the Discussion, especially because no effects of menopause on primary variables of interest such as PON1 and NHcy protein were observed, possibly because the number of premenopausal subjects was relatively small. Also, because menopause had no effect on PON1 and N Hcy protein we did not include the menopausal status into multiple regression analysis.
Comment 3
In the list of references, it is necessary to indicate DOI for many sources.
Response
We included DOI for all references for which they are available.
All changes made in the text are marked in red font in the revised manuscript. Additional references (7-9 and 55-58) are included in the reference list to support discussion suggested by the reviewers. The numbering of other references was updated accordingly in the text and the reference list. Finally, two additional supplementary tables (S4 and S5) were added to present results requested by one of the referees. In all responses we refer to line numbering in the revised manuscript.
Reviewer 3 Report
The Authors report elevated protein N-homocysteinylation in patients with RA that correlates with lower PON1. These findings are interesting and potentially relevant to the etiology/progression of the disease. However, the terminology used in the manuscript is confusing and a relevant background information is also missing. These deficiencies should be corrected as indicated below.
- Line 52-53: That PON1 hydrolyzes Hcy-thiolactone was originally established in studies with purified PON1 and then substantiated demonstrated in vivo in mice and humans.
- Line 54-55: Pon1-KO mice are also more sensitive to Hcy-thiolactone neurotoxicity, which should be pointed out.
- Lines 127-131 and Figure 1b: Referring to AA, AB, BB phenotypes may be of historic interest on the PON1 field, but its distracting in the present manuscript. Please use the modern PON1 genotype symbols instead.
- Lines 20, 21, 23, 26, 29: ‘protein bound Hcy thiolactone’ is incorrect and should be replaced by ‘N-Hcy-protein’. When Hcy-thiolactone reacts with a protein it generates N-Hcy-protein, not ‘protein bound Hcy thiolactone’.
- Line 154: the term ‘Hcy thiolactone bound to serum proteins’ should be replaced by ‘N-Hcy-protein’.
- Lines 221-224, Figure 1, lines 226, 238, 239, 241, 246 and 253: The use of term ‘Hcy thiolactone’ in this section is not correct. The term ‘protein bound Hcy thiolactone’ is incorrect here and throughout the manuscript, including Discussion, is also incorrect and should be replaced by the correct term ‘N-Hcy-protein’.
- Line 363 and Table 3: ‘protein bound Hcy thiolactone’ is incorrect and should be replaced by ‘N-Hcy-protein’.
- Line 374 and Table 3: ‘Hcy thiolactone’ should be replaced by ‘N-Hcy-protein’.
Author Response
Comment 1
Line 52-53: That PON1 hydrolyzes Hcy-thiolactone was originally established in studies with purified PON1 and then substantiated demonstrated in vivo in mice and humans.
Response:
The text has been changed to: “In addition, PON1 hydrolyzes homocysteine thiolactone both in vitro and in vivo” and the respective original studies are cited.
Comment 2
Line 54-55: Pon1-KO mice are also more sensitive to Hcy-thiolactone neurotoxicity, which should be pointed out.
Response:
The sentence has been modified to: “PON1 knockout mice are more sensitive to atherosclerosis and Hcy thiolactone-induced neurotoxicity” and the respective original study is cited
Comment 3
Lines 127-131 and Figure 1b: Referring to AA, AB, BB phenotypes may be of historic interest on the PON1 field, but its distracting in the present manuscript. Please use the modern PON1 genotype symbols instead.
Response
Nomenclature of PON1 polymorphism has been changed to RR, QR and QQ in the text, Figure 1b and tables.
Comment 4-8
Lines 20, 21, 23, 26, 29: ‘protein bound Hcy thiolactone’ is incorrect and should be replaced by ‘N-Hcy-protein’. When Hcy-thiolactone reacts with a protein it generates N-Hcy-protein, not ‘protein bound Hcy thiolactone’.
Line 154: the term ‘Hcy thiolactone bound to serum proteins’ should be replaced by ‘N-Hcy-protein’.
Lines 221-224, Figure 1, lines 226, 238, 239, 241, 246 and 253: The use of term ‘Hcy thiolactone’ in this section is not correct. The term ‘protein bound Hcy thiolactone’ is incorrect here and throughout the manuscript, including Discussion, is also incorrect and should be replaced by the correct term ‘N-Hcy-protein’.
Line 363 and Table 3: ‘protein bound Hcy thiolactone’ is incorrect and should be replaced by ‘N-Hcy-protein’.
Line 374 and Table 3: ‘Hcy thiolactone’ should be replaced by ‘N-Hcy-protein’.
Response to comments 5-8
“Protein-bound Hcy thiolactone” was replaced by “N-Hcy-protein” throughout the manuscript (text, tables, figures and legends).
All changes made in the text are marked in red font in the revised manuscript. Additional references (7-9 and 55-58) are included in the reference list to support discussion suggested by the reviewers. The numbering of other references was updated accordingly in the text and the reference list. Finally, two additional supplementary tables (S4 and S5) were added to present results requested by one of the referees. In all responses we refer to line numbering in the revised manuscript.
Round 2
Reviewer 2 Report
The article has been greatly improved. The relevance and novelty of the results obtained make the article very interesting for researchers. I have no more comments